# Young Muslim Perceptions of Their Socio-Educational Inclusion, Religiosity, and Discrimination in Spain: Identifying Risks for Understanding

**María Navarro-Granados** and **Verónica C. Cobano-Delgado Palma** *

Department of Educational Sciences, University of Extremadura, 06006 Badajoz, Spain; mariang@unex.es
* Correspondence: cobano@us.es

**Abstract:** The Muslim population is one of the religious groups facing the greatest obstacles to full socio-educational inclusion in the West. These are particularly noticeable among young people in areas such as access to employment. The purpose of this study was to find out their own perceptions of their socio-educational inclusion, discrimination, and religiosity. An eminently quantitative methodology was used, with an ad hoc questionnaire administered to a representative sample of a total of 1157 Muslims aged between 18 and 24. The results show that a higher level of religiosity is not related to a lower sense of belonging to Spanish society and should no longer be considered an obstacle to the socio-educational inclusion of young Muslims in Spanish society. On the other hand, their responses show that there is a relationship with greater perceived discrimination, especially in access to employment. In particular, women wearing hijab are substantially vulnerable. Young people, and especially Muslim women, make up a vulnerable population that requires specific school-to-work transition policies to improve their inclusion in the Spanish labour market. This research contributes to an important reflection based on the opinions of young Muslims themselves about supporting better socio-educational inclusion in Spain.

**Keywords:** education; social exclusion; Muslim

## 1. Introduction

Islam is the fastest-growing minority religion in Europe. In turn, Muslims form a sociological minority for three main reasons (Torrekens and Jacobs 2016): A. Numerically, they represent a small percentage of the national population; B. They tend to have a disadvantaged socio-economic position, as many Muslim immigrants were recruited as cheap labour in the 1960s and 1970s; C. Islam is not always on an equal footing with other religions, and, moreover, Islamophobia seems to be on the rise.

Although in recent years there has been a shift in EU integration policies towards a more multicultural stance, there is still popular resistance pushing for an assimilationist paradigm (Connor 2010). This has been exacerbated in the wake of terrorist attacks in the name of Islam, with the emergence of political parties with overtly Islamophobic discourse espousing this paradigm (Statham and Tillie 2016).

The socio-economic inclusion of young immigrants in Europe has become a major policy concern given high unemployment rates (Kogan et al. 2020). In terms of access to the labour market, authors such as Roth (2020) detect how immigrants, and more specifically those from Muslim-majority countries, face greater obstacles than natives in this regard.

The Muslim population is one of the most discriminated against religious groups in the West (Balkaya et al. 2019). This discrimination is especially notable among young people in areas such as access to employment (Di Stasio et al. 2019), with women who wear the hijab or other types of Islamic veil being especially vulnerable, currently presenting greater risk factors and exclusion in their labour market insertion (Weichselbaumer 2020). In the Spanish context, studies such as that of Fernández-Reino et al. (2023) show that the

level of discrimination against Muslim women wearing hijab when applying for a job is much lower than in other countries such as Germany and the Netherlands.

However, in any inclusion process, it is equally necessary to take into consideration the attitude of the minority population, in this case, the Muslim community, towards their inclusion in the host society (Norris and Inglehart 2012). In recent years, studies have been carried out that analyse the Muslim community's opinion on the compatibility between Islamic and Western values (Ciciora 2010). Of particular concern here are certain attitudes towards religious fundamentalism (Koopmans 2015) and gender equality (Diehl et al. 2009) that may hinder the successful inclusion of this community in the West.

The religiosity of Muslims has been at the centre of the debate on their inclusion in recent years (Torrekens et al. 2023). Indeed, it is often presented as an obstacle to the successful inclusion of this community in the West (Di Stasio et al. 2019). This is based on the false belief that Islam is incompatible with Western societies and their mainstream culture and values (Güngör et al. 2011; Leszczensky et al. 2020).

There are two main hypotheses about the religiosity of Muslims in the West (Fleischmann and Phalet 2012): A. The religious vitality hypothesis; and B. The secularisation hypothesis. According to the first hypothesis, there would be a maintenance of religiosity levels among the second generation, while for the second hypothesis, religiosity levels will be lower among the second generation as they are more integrated from a social, economic, educational, etc. point of view.

Research on this subject is not conclusive. Thus, some, such as Phalet et al. (2008), detect a growing secularisation among second-generation and more educated Muslims in the Netherlands. Maliepaard et al. (2012), for their part, describe a religious resurgence among the second generation in the Netherlands, indicating as a possible explanation that the ethnic segregation of neighbourhoods leads to a stronger attachment to the ethno-religious group.

Two of the most studied factors in relation to religiosity are feelings of national identity or belonging and discrimination. Several studies have shown that members of minority groups who perceive discrimination identify less with their nations (Badea et al. 2011; Jasinskaja-Lahti et al. 2008; Maxwell 2009; Verkuyten and Yildiz 2007). Leszczensky et al. (2020), in a study of Muslim adolescents of immigrant backgrounds in England, Germany, the Netherlands, and Sweden, find that discrimination is a stronger predictor of a low sense of national identity than religiosity. Verkuyten and Yildiz (2007), on the other hand, found with Turkish-Dutch Muslims in the Netherlands that perceived group rejection was associated with lower identification with Dutch society and higher identification with the ethnic minority, known as reactive ethnicity/identity. In this, members of the ethnic minority respond to rejection by reasserting their identity and withdrawing further into their minority group.

With regard to reactive religiosity, the research results are not unanimous. Some do not validate the hypothesis of reactive religiosity (Fleischmann and Phalet 2012; Torrekens and Jacobs 2016), nor do they detect a relationship between religiosity and perceived discrimination (Yazdiha 2018). Others, such as Connor (2010), find that in European countries where there is greater hostility towards Islam and its believers, they have higher levels of religiosity in response. Along the same lines, what Maliepaard and Verkuyten (2018) identify as reactive identity stands out. In this sense, perceived group discrimination (feeling that your group is being excluded and unfairly treated) is a factor that predicts a low sense of national belonging, leading to distancing from the majority group. This may explain, according to the European Union Agency for Fundamental Rights (FRA 2018), why many second-generation Muslims feel less attached to the host society than immigrants.

However, it is important to bear in mind a number of considerations about inclusion, specifically the feeling of belonging to the host society. As Hur (2022) rightly points out, this is an interactive process of constant negotiation, which depends on both the attitude of the immigrants and the intention of the host society to include them.

Spain represents, both for its geographical position and the historical legacy of Al-Andalus, the most evident link with Islam in the European Union. The Muslim heritage is undeniable in language, culture, architecture, etc. (Astor 2017). However, attempts have often been made to erase its Muslim past (Rzepnikowska 2023), attempting to construct Spain's national identity by eliminating its Muslim roots. In fact, it is argued that the current stereotypes associated with Muslims in Spain are not disconnected from their historical past (Manzano 2014).

The Spanish State is defined in its Constitution as a non-confessional state that maintains cooperative relations with the different religious confessions (Spanish Constitution 1978, art. 16). Despite having a Cooperation Agreement with the Islamic Commission of Spain that protects the rights of this community in our country, many of them are still not properly covered, such as the lack of mosques and Islamic cemeteries (Torrens 2019).

To our knowledge, there are no studies in Spain that address the Muslim community's perception of their religiosity and inclusion using a quantitative methodology. Moreover, we complement previous research that has only focused on Muslim immigrants (Maliepaard and Alba 2016; Scheible and Fleischmann 2013; Martinovic and Verkuyten 2012; Van Bergen et al. 2016) by including second-generation Muslims and converts in the sample.

In this study, we address the following central research question: What are the perceptions of young Muslims in Spain about their religiosity, socio-educational inclusion, and discrimination? We also seek to answer the following specific research questions: What factors explain greater religiosity and discrimination among respondents; is there a relationship between the degree of religiosity and perceived discrimination; what factors explain a lower sense of belonging to Spanish society; is there a relationship between discrimination in access to employment and the feeling of belonging to Spanish society; and is there a relationship between discrimination in access to employment and the feeling of belonging to Spanish society?

Consequently, the aim of this article is to contribute to the discussion and understanding of the socio-educational inclusion of young Muslims in a society that is often polarised on issues of immigration and identities. It also discusses the implications of these results for an appropriate school-to-work transition.

## 2. Materials and Methods

### 2.1. Participants

According to the demographic study of the Muslim community in Spain carried out by UCIDE (2020), a total of 2,091,656 Muslims were registered on that date. As this number is higher than 10,000 subjects, to calculate the sample, we used the formula established for statistically infinite populations (>10,000 subjects): n = z2 P*Q/E2. With a confidence level of 97% and a margin of error of 3%, we obtained a sample of 1157 Muslims. The inclusion requirements for our sample were self-identification as Muslim (more or less practicing), residing in Spain, and being between 18 and 24 years old. We go beyond those who take being an immigrant as an inclusion criterion, thus failing to reach second generations and converts (Maliepaard and Alba 2016; Scheible and Fleischmann 2013; Martinovic and Verkuyten 2012; Van Bergen et al. 2016). These are very specific criteria that are not reflected in the available demographic studies, so the sample was selected through non-probabilistic sampling.

As a sampling technique, following Pruchno et al. (2008), we used quota sampling, which is the most widespread in social surveys. In this way, we tried to partially mitigate the limitations of non-probability sampling in terms of representativeness and generalisability of the results. To do so, we kept proportionality between the data available in the demographic census used (UCIDE 2020): foreign Muslims and Spaniards for each autonomous community (a territorial entity that, within the current constitutional legal system in Spain, is endowed with autonomy in certain competences). Specifically, we

selected Catalonia, Madrid, and Andalusia, the three with the largest Muslim population in Spain.

According to the census, we found in Madrid a total of 118,740 foreign Muslims and 180,571 Spaniards; in Catalonia, 359,883 foreigners and 204,172 Spaniards; and in Andalusia, 191,391 foreigners and 149,678 Spaniards. Proportionality was maintained in the selection of participants for inclusion in the final sample. Thus, the final sample shown in Figure 1 was obtained.

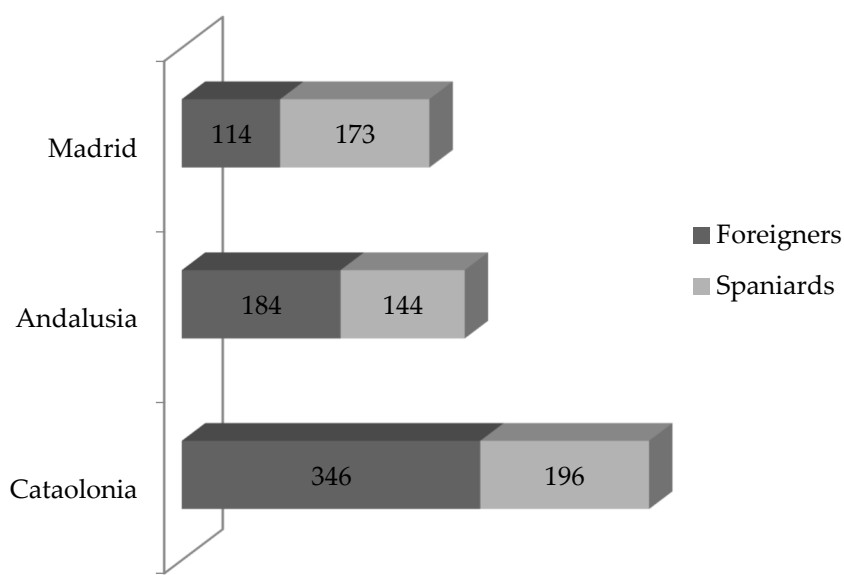

**Figure 1.** Number of foreign and Spanish Muslims by autonomous community included in the sample. Source: own elaboration.

To access the study sample, a list of mosques, associations, and Islamic educational centres was drawn up for each autonomous community. We used different techniques similar to those of other research that also employ non-probability sampling (Brummett et al. 2006; Pruchno et al. 2008): word of mouth and posts on social networks (Facebook, Instagram, and Twitter) by our research group and by the organisations and mosques that wanted to collaborate. To obtain a greater number of responses, the questionnaires were passed both on paper and online through Google Forms.

These were translated into French, English, classical Arabic, and the Moroccan dialect (Dariya), as the Muslim population in Spain is mainly Moroccan. It was informed that participation was completely voluntary and that their answers would be confidential.

The sample was made up of 42.7% men and 57.3% women. In addition, 57.5% are immigrants, 33.4% are second-generation, and 9.1% are converts. The majority of the respondents were born in Spain (43.3%) and Morocco (38.8%). Also, 39.7% had higher education (university), 39.2% had secondary education (baccalaureate, vocational training, etc.), 17.9% had primary education, and 3.3% had no education. In addition, 54.6% are students, 31.4% are working, 8.6% are unemployed, 0.2% are retired, and 5.3% are housewives. The majority (73.9%) say they feel well off financially, 13.9% poorly off, 9.5% very well off, and 2.7% very poorly off. In addition, 96.1% practice Sunni Islam, and 3.9% practice Shi'a Islam.

### 2.2. Instruments

A questionnaire is used that was developed ad hoc for a broader research project entitled "peace and non-violence in the teaching of Islam in Spain: socio-educational aspects". It is made up of a first block of identifying data and a total of eight dimensions:

1. Religiosity;
2. Religious beliefs;

3. Islamic teaching;
4. Socio-educational inclusion;
5. Discrimination;
6. Group injustice;
7. Preventive socio-educational measures;
8. Views on Islamic-inspired terrorism.

In this study, we focus on religiosity, religious beliefs, socio-educational inclusion, discrimination, and group injustice.

Reliability was calculated using Cronbach's alpha coefficient, obtaining a value that indicates good reliability (0.763). To analyse construct validity, two procedures were carried out:

A  Exploratory factor analysis (EFA) using the principal components method. Previously, sample adequacy was verified with the Kaiser-Meyer-Olkin (K-S) test and Bartlett's sphericity test. By means of the Varimax rotation method, the interpretation of the dimensions that make up the questionnaire was refined as much as possible. All of them reached adequate measures of sampling adequacy, saturating the items in each factor with a value higher than 0.4 (Floyd and Widaman 1995).

B  Non-Metric Multidimensional Scaling, or PROXSCAL. The goodness-of-fit of the model was confirmed, with stress values (measuring data mismatches) close to 0 and measures of fit (DAF and Tucker) close to 1 (Biencinto et al. 2013).

*2.3. Measures*

We differentiate between core and secondary measures. The core measures of the present research are the following: A. religiosity; B. socio-educational inclusion; and C. discrimination.

The secondary measures are: religious beliefs; perceived group injustice; generation (immigrants, second generations, and converts); length of residence in Spain; educational level; gender; and whether they wear hijab or another type of headscarf.

2.3.1. Religiosity

This dimension of the questionnaire, which forms a central variable in our study, measures the respondents' level of religiosity. To this end, and in line with other authors such as Maliepaard and Phalet (2012) and Leszczensky and Pink (2020), subjective religiosity was differentiated from practical religiosity. Thus, the scale measured: A. Personal importance attached to Islam in their lives; B. Subjective religiosity (subjective religiosity). Subjective religiosity (the extent to which they consider themselves religious) (Guveli 2015); and C. Religious practices (prayer, Qur'an reading, and mosque attendance). A Likert-type scale with five response options (1 = not at all and 5 = very much) was used.

The items that make up the questionnaire are the following:

-  Do you consider yourself more or less religious than your parents? More religious, less religious and the same.
-  Religion is very important in my life.
-  I consider myself a religious person.
-  I do Salat.
-  I go to the mosque.
-  I read the Koran.

2.3.2. Religious Beliefs

According to the factor analysis carried out, this variable formed a dimension or construct different from that of religiosity. In this dimension, we measured the opinion and position of respondents on a series of Islamic beliefs. In the present study, we focused specifically on the degree of agreement with literalist interpretations of sacred texts.

A Likert-type scale with four response options (1 = strongly disagree and 4 = strongly agree) was used. The items that make up the scale are the following:

- I believe that a literal interpretation of sacred texts should be made.
- Sacred texts should be interpreted in a way that is adapted to the times in which we live.
- Religious practices should be adapted to the social norms of the country in which we live.
- The wearing of the veil by Muslim women should be optional.
- For Muslim women to wear the veil is an act of obedience to Allah.

### 2.3.3. Socio-Educational Inclusion and a Sense of Belonging

For authors such as Leszczensky et al. (2020), national identification is a key aspect of social identity and indicates a sense of belonging to the country. It is also a key component of minority inclusion.

In our study, we were interested in subjective feelings of belonging, i.e., to what extent they feel Spanish, from their country of origin or their parents' country, from both or neither. Studies such as Giuliani et al. (2018) and Verkuyten and Yildiz (2007) distinguish between national identification (in this case, feeling Spanish) and religious identification (feeling Muslim), making them choose between one or the other. In the present study, this division is not shared since Muslims and Spanish are totally compatible.

It was evaluated with the following item: 'How do you feel more?' A. Spanish; B. Of your country of origin or of your parents; C. Of both; D. Of neither.

The rest of the scale is composed of a total of six items with a Likert-type scale of four response options (1 = strongly disagree and 4 = strongly agree):

- I feel that in Spain I can maintain my religious identity and customs.
- I feel that I have had to reject/put aside some values/customs of my religion in order to feel more integrated in Spain.
- In Spain, being or feeling integrated is confused with having to put aside our religious beliefs and customs.
- In my daily life I interact with non-Muslims.
- I know Muslim women who have thought about removing their veils to feel more integrated in Spain.
- I know Muslim women who have thought about wearing the veil to defend/claim their religious identity in Spain.

### 2.3.4. Perceived Personal Discrimination

As in Fleischmann and Phalet's (2012) study, it was found through the construct validity of the questionnaire that personal discrimination and perceived group injustice are two different constructs.

Regarding personal discrimination, it is worth noting that the present study analyses a subjective phenomenon that, as Steinmann (2019) argues, may differ from actual discrimination. However, it is very difficult to differentiate between what is perceived as a discriminatory act and whether it really is one, regardless of what the person perceives as such.

A distinction was made between discrimination based on ethnicity and discrimination based on religion, and the scale consisted of a total of five items with a Likert-type scale of four response options (1 = strongly disagree and 4 = strongly agree) and a multiple-choice item. In the latter, they had to indicate in which aspects they thought they had received unfavourable treatment for being Muslim in Spain: education, employment, construction of mosques, access to housing, and others.

- In recent years I have felt discriminated against in Spain for being a Muslim.
- In recent years I have felt discriminated against in Spain because of my race/ethnicity/skin colour.

- I think it is difficult to be a Muslim in Spain.
- Most Spaniards (non-Muslims) consider me a foreigner.
- I have started to practice my religion more discreetly for fear of being associated with terrorism.

### 2.3.5. Perceived Group Injustice

As Yazdiha (2018) argues, people do not need to have a personal experience of discrimination to feel that their group is being discriminated against. This scale is composed of a total of three items with a Likert-type scale of four response options (1 = strongly disagree and 4 = strongly agree).

- In Spain, people of other religions are treated better than Muslims.
- In Spain, Muslim immigrants are less well regarded than other immigrant groups.
- In Spain, a Muslim is less likely to be called for a job interview than a non-Muslim.

### 2.4. Data Collection and Analysis Procedure

The following data analysis techniques are used:

A    Descriptive statistics: percentages, mean ($\bar{x}$), and standard deviation ($\sigma$).
B    Correlations (contingency coefficient for nominal variables and Pearson's correlation coefficient for ordinal variables).
C    Inferential statistics. In reference to the latter, non-parametric tests are used, specifically the Kruskal-Wallis H test and the Mann-Whitney U test, once the non-normality of the sample has been verified by means of the Kolgomorov–Smirnov (K-S) test ($p < 0.05$).

SPSS v. 27 was used as the quantitative analysis programme.

## 3. Results

### 3.1. Socio-Educational Inclusion (Feeling of Belonging)

Regarding the feeling of belonging, the majority (62.8%) indicate that they feel that they belong to their country of origin or that of their parents, followed by 22.6% who indicate the option "both" (both to their country of origin or that of their parents and to Spain). Table 1 below shows the statistically significant differences obtained for the variable "feeling of belonging".

**Table 1.** Significant differences for the variable "feeling of belonging".

| | H Kruskal-Wallis | Significance Level ($p$) | Mean Rank |
|---|---|---|---|
| I believe in a literal interpretation of the sacred texts (religious belief scale) | 1156.000 | 0.000 | None (1135.00) Spanish (62.50) |
| In my daily life I interact with non-Muslims (socio-educational inclusion scale) | 749.583 | 0.000 | Spanish (1095.50) None (80.00) |
| I consider that in Spain I can maintain my identity and customs as a Muslim (socio-educational inclusion scale) | 427.694 | 0.000 | Both (795.00) Neither (59.50) |
| I feel that I have had to put aside the values/customs of my religion in order to feel integrated in Spain (socio-educational inclusion scale) | 1135.00 | 0.000 | None (1100.00) Spanish (60.02) |
| Scale Personal discrimination | 214.532 | 0.000 | None (983.00) Spanish (306.50) |
| Scale Group injustice | 263.020 | 0.000 | None (969.00) Spanish (262.00) |

Source: own elaboration.

With an H value of 1156.000 and an optimal level of significance ($p$ = 0.000), those in favour of literalist interpretations feel to a greater extent that they do not belong either to their country of origin or that of their parents, or to Spain (mean rank = 1135.00), as opposed to those who disagree to a greater extent and who indicate to a greater extent that they feel Spanish (mean rank = 62.50). No differences are detected according to the degree of religiosity.

Respondents who say that they interact with non-Muslims on a daily basis are more likely to feel Spanish than those who do not (H = 749.583; $p$ = 0.000; average range: 1095.50 vs. 80.00). Equally interesting is that those who consider that in Spain they can maintain their religious identity and customs indicate to a greater extent that they belong to both (both to their country of origin or their parents' country and to Spain), and those who do not agree select the option 'neither' (H = 427.694; $p$ = 0.000; average range 795.00 versus 59.50). Likewise, those who say that they have had to reject or set aside customs and values of their religion in order to feel more integrated in Spain indicate to a greater extent that they belong to none, compared to those who have not had to do so and who select the option "Spanish" (H = 1156.000; $p$ = 0.000; mean range: 1135.00 vs. 62.50).

Muslims who perceive greater personal discrimination and group injustice indicate to a greater extent the option 'none' (average ranges = 983.00; 969.00), compared to those who report lower levels, who feel more Spanish (306.50; 262.00). Moreover, those who report having suffered unfavourable and/or discriminatory treatment because of being Muslim in access to employment also report feeling that they do not belong to Spain, their country of origin, or their parents' country (Contingency Coefficient = 0.810; $p$ = 0.000).

Finally, with regard to the feeling of belonging, through the contingency coefficient (0.782; $p$ = 0.000), we detected a statistically significant relationship with the variable "generation". It is particularly striking that a significant percentage of second-generation Muslims feel that they belong neither to their country of origin or that of their parents nor to Spain.

### 3.2. Socio-Educational Inclusion

A considerable percentage of the young Muslims surveyed say that they interact with non-Muslims in their daily lives (68.5%). Statistically significant differences are detected with the variables 'length of residence in Spain', 'generation', 'religiosity', 'personal discrimination', and 'group injustice'. In this way, they claim to relate to non-Muslims to a greater extent.

- Those who have been residing in Spain longer (H = 37.355; $p$ = 0.000) versus those who have been residing less (mean range = 606.11 and 400.00).
- Muslim converts (H = 492.195 and $p$ = 0.000) versus immigrants (mean range = 1095.50 and 515.72).
- Less religious Muslims (H = 336.116 and $p$ = 0.000) versus more religious Muslims (mean rank = 961.30 and 460.92).
- Those who perceive less personal discrimination (H = 90.874 and $p$ = 0.000) versus those who perceive more (mean rank = 942.09 and 469.57).
- Those who perceive less group injustice (H = 99.508 and $p$ = 0.000) versus those who perceive more (mean range = 816.74 and 527.42).

In addition, 74.6% consider that in Spain they can maintain their identity and customs as Muslims. However, a non-insignificant percentage (66.7%) say that they have had to reject or put aside some values and customs of their religion in order to feel more integrated in Spain. Likewise, a high percentage (82.2%) consider that being or feeling integrated in Spain is confused with having to put aside their customs and religious beliefs.

### 3.3. Religiosity

Additionally, 38.1% consider themselves more religious than their parents, 35.9% are the same, and 26% claim to be less religious. On a scale of 1 to 5 (1 = not at all and 5 = very

much), the following descriptive statistics are obtained regarding the level of religiosity (Table 2):

**Table 2.** Descriptive statistics (mean and standard deviation) for the variable "religiosity".

|  | Media | Standard Deviation |
|---|---|---|
| Religion is very important in my life | 4.41 | 1.007 |
| I consider myself a religious person | 3.72 | 1.211 |
| I do Salat | 3.20 | 1.042 |
| I go to the mosque | 3.98 | 1.197 |
| I read the Qur'an | 3.87 | 1.242 |

Source: own elaboration.

We obtain higher levels of religiosity among women (U = 40,904.000; $p$ = 0.000; mean rank = 759.12 vs. 328.98) and, more specifically, among those who wear hijab or other Islamic headscarves (U = 42,096.500), and also among those who perceive greater personal discrimination (H = 285.279; $p$ = 0.000; mean range = 752.76 vs. 282.25). In addition, we found a significant correlation ($p$ = 0.000), albeit moderate ($r_{xv}$ = 0.576), between the variables "religiosity" and "perceived personal discrimination". Thus, those with higher levels of religiosity also perceive greater discrimination, and vice versa. No differences were found between religiosity and perceived group injustice.

With respect to generation, the following differences are detected (Table 3):

**Table 3.** Significant differences for the variable "generation".

|  | H Kruskal-Wallis | Significance Level ($p$) | Mean Rank |
|---|---|---|---|
| Religion is very important in my life | 0.258 | 0.879 | -- |
| I consider myself a religious person | 315,620 | 0.000 | Converts (913.75) Second generation (368.24) |
| I do Salat | 76,379 | 0.000 | Second generation (668.58) Immigrant (511.40) |
| I go to the mosque | 76,464 | 0.000 | Second generation (682.49) Immigration (515.73) |
| I read the Qur'an | 63,152 | 0.000 | Second generation (672.43) Immigrant (519.58) |

Source: own elaboration.

As can be seen, no generational differences are obtained for the item "religion is very important in my life". Differences were found for subjective religiosity (considering oneself a religious person) and religious practices. In the former, converts consider themselves more religious than second-generation Muslims, and second-generation Muslims have higher levels of religious practice than immigrants. However, the differences in religious practices are small.

With respect to educational level, differences with an optimal level of significance are obtained in all cases ($p$ = 0.000), with higher levels of religiosity among those with a higher level of education.

*3.4. Discrimination*

According to the average calculated both for the dimension as a whole and for each of its component items (see Table 4), a considerable percentage of Muslims perceive personal discrimination, with the average of the scale close to value 3 (agree). The highest percentages correspond to religious discrimination and the feeling that the majority of Spaniards (non-Muslims) consider them to be foreigners.

**Table 4.** Descriptive statistics for the dimension 'perceived personal discrimination'.

| Dimension/Item | Percentage | x̄ | σ= |
|---|---|---|---|
| 1. In recent years I have felt discriminated against in Spain because I am a Muslim. | | 2.9 | 0.92 |
| 2. In recent years I have felt discriminated against in Spain because of my race/ethnicity/skin colour. | 67% | 3.03 | 1.102 |
| 3. I think it is difficult to be a Muslim in Spain. | 55.3% | 2.88 | 1.153 |
| 4. Most Spaniards (non-Muslims) consider me a foreigner. | 54% | 2.88 | 1.128 |
| 5. I have started to practise my religion more discreetly for fear of being associated with terrorism. | 68.5% | 3.17 | 1.063 |

Source: own elaboration.

With the dimension mean, they perceive more personal discrimination, with an optimal level of significance.

- Muslims belonging to the second generation vs. immigrants (H = 114.196; $p$ = 0.000; mean rank = 716.33 vs. 492.27).
- Women vs. men (U = 71,704.500; $p$ = 0.000; mean rank = 711.19 vs. 391.44).
- Women wearing hijab vs. women not wearing hijab (U = 19,128.500; $p$ = 0.000; mean rank = 430.72 vs. 215.42).

Regarding the question 'indicate if you have received or receive unfavourable treatment for being Muslim in the following aspects in Spain' (construction of mosques, education, employment, access to housing, etc.), the highest percentage corresponds to access to employment (44.4%), followed by access to housing (20.9%). In both cases, the results are significantly higher among women, especially among those wearing hijab (contingency coefficient = 0.822; $p$ = 0.000).

## 4. Discussion and Conclusions

Through this study, we have sought to find out the opinions of young Muslims on their socio-educational inclusion, religiosity, and possible discrimination in Spain.

With regard to their socio-educational inclusion, more specifically in relation to the feeling of belonging to Spanish society, we detected greater feelings of uprootedness (i.e., they feel that they do not belong to Spain or to their country of origin or that of their parents) among young Muslims who are in favour of literalist interpretations of the Koran. We did not detect differences according to the respondents' level of religiosity. This is an important finding in order to stop considering the level of religiosity of young Muslims as an obstacle to their successful inclusion in the host society; in this case, Spain. Our results are not in line with previous studies that found a lower sense of belonging to the host society among the most religious Muslims (Fleischmann and Phalet 2012; Leszczensky et al. 2020).

Greater feelings of uprootedness are also detected among young people who report suffering processes of cultural assimilation (they feel they have to put aside the values and customs of their religion in order to feel more integrated in Spain) and among those who perceive higher levels of discrimination and injustice towards Muslims. More specifically, we found that suffering unfavourable and/or discriminatory treatment in access to employment is a factor related to a lower sense of belonging to Spanish society. In this line, our results are consistent with previous studies that indicate how discrimination is a predictor of a low sense of belonging to the host society (Leszczensky et al. 2020). In relation to these results, the thesis of Balkaya et al. (2019) makes sense when they indicate that perceived unfair treatment by the majority group may lead minorities to distance themselves from what they see as a source of distress.

Finally, regarding the feeling of belonging, young people belonging to the second generation and those who do not interact with non-Muslims in their daily lives show a lower feeling of belonging to Spanish society. In this line, our results do not coincide with

those of Beek and Fleischmann (2020), who found no differences according to generation. By detecting that a significant percentage of this generational group feels that they do not belong either to Spain or to their country of origin or that of their parents, we can glimpse possible problems of uprooting that, without a doubt, may pose a serious problem for the successful inclusion of this group.

The results obtained have direct implications for the socio-educational measures proposed to improve the sense of belonging and rootedness of young Muslims in Spain. According to Spiegler et al. (2022), these processes of identity construction are influenced by the degree to which they perceive belonging to the host society as harmonious vs. conflictive. In this sense, discourses that question the compatibility of Islam with Western ways of life are worrying (Kranendonk et al. 2018), discourses that only aggravate these feelings of uprootedness.

We are aware that feelings of belonging and uprootedness are influenced by multiple explanatory variables that are difficult to analyse in a quantitative study. For this reason, we believe that it would be interesting to carry out future research that explores this issue in greater depth using a qualitative methodology. Specifically, we could analyse the extent to which young Muslims are developing a multiple identity strategy that recognises both their ethno-religious origin and their belonging to the host society, as has been performed in previous studies in other countries such as Belgium (Torrekens et al. 2023).

In terms of religiosity, we find higher levels among women (more specifically, among those who wear hijab). Also noteworthy is the bidirectional relationship between religiosity and perceived personal discrimination. Thus, young Muslims with higher levels of religiosity also report higher levels of discrimination, and vice versa. Along these lines, it would be advisable to explore in greater depth in future studies whether the phenomenon known as 'reactive religiosity' is occurring in Spain, whereby members of the ethnic/religious minority respond to rejection and discrimination by withdrawing more into their minority group and thereby increasing their religiosity (Verkuyten and Yildiz 2007).

Unlike Beek and Fleischmann (2020), in our study we detect higher levels of religious practice among second-generation Muslim youth. In this case, we do not agree with those authors who envisage a greater assimilation in religious terms among the new generations (Checa and Monserrat 2015). This could be related to the alarming levels of discrimination and injustice reported by the young Muslims surveyed, which could lead many of them to react by increasing their own religiosity (Voas and Fleischmann 2012).

The Muslim population is one of the most discriminated religious groups in the West (Balkaya et al. 2019), and their socio-economic inclusion is a major concern given high unemployment rates (Kogan et al. 2020). In terms of access to the labour market, authors such as Roth (2020) find that immigrants, and more specifically, those from Muslim-majority countries, face greater obstacles than natives. According to the results obtained, we are faced with a generation of young people who, despite having been born in Spain, continue to suffer discrimination in different aspects of their lives, especially in access to employment.

The high levels of perceived personal discrimination and group injustice detected in this study lead us to reflect on the road that remains to be travelled to achieve true inclusion of young Muslims in Spain.

In the results obtained, there is an important gender dimension.

We found that the discrimination levels are particularly noticeable among women (more specifically, among those who wear hijab) in areas such as access to employment. In this sense, this group presents greater risk and exclusion factors in their labour market insertion (Weichselbaumer 2020). It is, therefore, a vulnerable group that requires specific school-to-work transition policies (a period that is particularly sensitive to the inclusion/exclusion of young people) that can improve their inclusion in the labour market (Hein 2012). Thus, previous research shows that Muslim women are offered fewer career opportunities, as well as a bias towards certain lower-paid job categories (Mumtaz and

Awais 2022). It would be advisable to implement social and educational policies aimed at reducing gender inequality in this transition (Beck et al. 2006).

The results of our research reaffirm the findings obtained in previous research, which concluded that Muslim women are particularly vulnerable to discrimination due to, among other issues, religious 'visibility' in the case of those who wear the hijab or other veils (Baboolal 2023; Weichselbaumer 2020). They are often victims of so-called gender Islamophobia, seen as submissive, backward, and oppressed (Iner et al. 2022). Although in the Spanish context, some studies such as Fernández-Reino et al. (2023) show that the level of discrimination against Muslim women wearing hijab when applying for a job is much lower than in other countries such as Germany and the Netherlands, our results highlight the need to carry out studies that focus on analysing in greater depth the unique experiences of discrimination faced by these women in the Spanish context. Along these lines, we consider it interesting that future studies delve, from an intersectional perspective (Khattab and Hussein 2018), into the obstacles and barriers faced by Muslim women in their access to the labour market in Spain. In this way, the multiple discriminations they suffer in this regard as women, Muslims, and, in many cases, racialised, will be revealed.

The assimilation processes reported by the young people surveyed are related to the erroneous belief, often widespread, that Islam is incompatible with Western values, so that these young people are required to fully adhere/assimilate to the latter (Checa and Monserrat 2015).

Understanding inclusion as a necessarily bidirectional process, we agree with Furió (2007) when he states that, when talking about the failure of inclusion policies for this group, the focus is almost exclusively on the disqualification of Islam and on the allegedly non-inclusive attitudes of Muslims, and not on the self-criticism of the system itself. In this sense, and given the high percentage of respondents who maintain that being integrated in Spain is confused with processes of assimilation, we consider it necessary to implement socio-educational measures aimed especially at the host society (autochthonous). To this end, it is essential to raise awareness in a discourse that goes against the unfortunate theory of the "clash of civilisations", to promote interreligious coexistence, to encourage mutual understanding, and to dismantle the ethnocentric discourse that points to Muslims as the "others" who do not want to integrate (Torrekens et al. 2023).

Finally, it is important to point out some limitations of the present study. The descriptive, correlational, and inferential design of the study limits establishing any possible causal relationship between the variables under study. We find that greater religiosity is related to perceived greater discrimination, but it is also plausible that greater perceived discrimination leads to greater religiosity among the young Muslim respondents. Future studies can explore possible causal relationships between inclusion, discrimination, and religiosity through experimental designs. Longitudinal studies analysing possible changes in these relationships would also be interesting.

**Author Contributions:** Conceptualization, M.N.-G. and V.C.C.-D.P.; methodology, M.N.-G.; software, M.N.-G.; validation, M.N.-G.; formal analysis, M.N.-G.; investigation, M.N.-G. and V.C.C.-D.P.; resources, V.C.C.-D.P.; data curation, M.N.-G.; writing—original draft preparation, M.N.-G. and V.C.C.-D.P.; writing—review and editing, M.N.-G. and V.C.C.-D.P.; visualization, M.N.-G. and V.C.C.-D.P.; supervision, M.N.-G. and V.C.C.-D.P.; project administration, V.C.C.-D.P.; funding acquisition, V.C.C.-D.P. All authors have read and agreed to the published version of the manuscript.

**Funding:** This research was funded by Research Plan of the University of Seville [VPPI-US-2016] and by Development and Cooperation Office of the University of Seville in the framework of the project 'Islam and peace through Muslim voices. Preventive socio-educational measures' [AYP/16/2018].

**Institutional Review Board Statement:** Ethical review and approval for this study was waived because the data collection instrument was designed in 2018, when the University of Seville did not have an official ethics committee for social science research or an established protocol. Participation in the research was voluntary and anonymous. The ethical requirement of informed consent was

followed, which restricts the use of the information obtained solely for research purposes and guarantees the anonymity and confidentiality of the participants.

**Informed Consent Statement:** The formed consent was obtained from all subjects involved in the study.

**Data Availability Statement:** The data presented in this study are available on request from the corresponding author. The data are not publicly available due to privacy concerns.

**Conflicts of Interest:** The authors declare no conflict of interest.

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
