# Peer review of "Young Muslim Perceptions of Their Socio-Educational Inclusion, Religiosity, and Discrimination in Spain: Identifying Risks for Understanding"

_socsci, doi:10.3390/socsci13030156_

Round 1

Reviewer 1 Report

Comments and Suggestions for Authors

This is a topic of interest and relevance, especially in these days when religious integralisms could tarnish quality education. For this reason, studies like this one are necessary in which the opinion, first-hand, of the main people involved is known. Especially from the socio-educational perspective and detection of possible discrimination, since this can help when it comes to improving, for example, the learning climate in classrooms, avoiding racial conflicts from the first years of schooling. The methodology and presentation of the work are correct, despite the limitations in correlations, as the authors themselves indicate. Greater incidence is suggested in the analysis from a gender perspective, since the authors indicate that there are differences by sex, but the results are not analyzed in greater depth, and it could be relevant for the interpretation of the results.

Author Response

LETTER TO REVIEWER 1:

Greater incidence is suggested in the analysis from a gender perspective, since the authors indicate that there are differences by sex, but the results are not analyzed in greater depth, and it could be relevant for the interpretation of the results.

  • RESPONSE:

In view of the word limit, we have elaborated on this issue in the discussion section: “We found that the discrimination levels is particularly noticeable among women…” lines 499-514.

Reviewer 2 Report

Comments and Suggestions for Authors

The topic of this article is very interesting, especially in the context of the growing challenges of coexistence among different religious and ethnic groups in Europe and the dynamic influx of new migrants. This article appears to be rigorous in terms of the conducted sociological research. Additionally, the obtained results and their interpretation are fundamentally correct and contextualized. This submission has the potential to inspire scholarly discussions and further research.

However, I have a suggestion for the authors to explicitly state in the title of this article that its subject is the subjective opinions of young Muslims living in Spain rather than objective criteria of discrimination. As Steinmann (2019) notes, these two issues can significantly differ from each other, and leaving the current title may be misleading.

Furthermore, in the first part, the author(s) rightly describe manifestations of the lack of integration of Muslims in Spain and Europe (e.g., in employment issues), but fundamentally, they attribute the blame for this problem solely to the receiving societies. This does not seem to be a fair approach, as it is a fact that many Muslims do not want to integrate and engage in work. Often, they choose to live in ethnic groups and rely on welfare. Numerous cases of this nature are frequently reported in the media across Europe. Despite the authors' denials, it also happens that some Muslims contest Western values. Moreover, they attempt to impose their values on the inhabitants of the receiving countries (e.g., attitudes toward alcohol, adherence to certain Sharia principles, etc.). It seems that in the introductory part, there is a need to balance the description of the current situation to make it more objective.

Author Response

LETTER TO REVIEWER 2:

However, I have a suggestion for the authors to explicitly state in the title of this article that its subject is the subjective opinions of young Muslims living in Spain rather than objective criteria of discrimination. As Steinmann (2019) notes, these two issues can significantly differ from each other, and leaving the current title may be misleading.

Furthermore, in the first part, the author(s) rightly describe manifestations of the lack of integration of Muslims in Spain and Europe (e.g., in employment issues), but fundamentally, they attribute the blame for this problem solely to the receiving societies. This does not seem to be a fair approach, as it is a fact that many Muslims do not want to integrate and engage in work. Often, they choose to live in ethnic groups and rely on welfare. Numerous cases of this nature are frequently reported in the media across Europe. Despite the authors' denials, it also happens that some Muslims contest Western values. Moreover, they attempt to impose their values on the inhabitants of the receiving countries (e.g., attitudes toward alcohol, adherence to certain Sharia principles, etc.). It seems that in the introductory part, there is a need to balance the description of the current situation to make it more objective.

RESPONSE:

  • We have included the word "perceptions" in the title.
  • We have included the paragraph in order to balance what the reviewer says: “However, in any inclusion process it is equally necessary to take into consideration the attitude of the minority population…” lines 50-57.

Reviewer 3 Report

Comments and Suggestions for Authors

The resistance towards an assimilationist policy by Muslim youth is key to your paper, as is the non-discriminatory access to labour markets, conflicting religiosity paradigms especially around identity/belonging  with second-gen youth vs immigrant youth,  Spain's  historic link with Islam being minimized, the tragedy that discrimination is creating a sense of isolation from Spain or the country of origin, even though 74.6% claim they can maintain their identity and customs and the most striking finding is that they don't have to put feel that they have to put aside these customs and beliefs in order to feel integrated, while 66% feel they have to....and that Spaniards still consider them as foreigners.

The questions of how employment and discrimination alienate youth warrants further more specific research with labour market (systems) analysis. Findings such as yours may not be able to put to rest how reactive religiosity plays out and how successful integration of Muslims/Islam generally is a challenge for Spain/the west. 

A great paper that deservedly contributes to the literature by delineating the findings with other conclusions in literature thus far. 

Author Response

He has not suggested any changes to us.

We are grateful for the many comments made.

Reviewer 4 Report

Comments and Suggestions for Authors

This article considers the relationship between religiosity and national belonging from the perspective of young Muslims in Spain. This is a significant topic of research given that mainstream notions of national belonging are often implicitly or explicitly based on ethno-religious exclusions. However, the research design for this study is unclear, especially in terms of definitions for its main variables and how relationships between variables was analyzed.

-- What is the central research question for the study? Lines 113-119 list a series of questions and it is not clear if there is an overarching question for the study, or if the study is a series of sub-questions examining relationships between a number of variables (religiosity, discrimination, belonging)? The term 'integration' is mentioned in lines 107-108, but is not part of the research questions.

-- It is mentioned that 'the study population consisted of a total of 2,091,656 Muslims' [line 127]. Is this the total number of Muslims between 18-24 years in Madrid, Andalusia, and Catalonia? It is then mentioned that 1,157 respondents were recruited for this study. This sample however is not a random sample since respondents were recruited through Muslim associations? The process of recruitment needs to be clarified further. It is fine if this is a convenience sample (which cannot be generalized to the wider population of Muslim youth in the three communities) since the study seeks to target respondents with specific characteristics.

-- The measures mentioned in Section 2.3 need to be clearly defined. Are the questions mentioned in Section 2.3 different from the questions listed in Tables 1-4? What is the difference between religiosity and religious beliefs? Are socio-educational inclusion and sense of belonging part of the same measure?

-- Is religiosity the central variable? Are correlations being measured in terms of the relationship of religiosity to the other measures in Section 2.3? Or is belonging the central variable? Or are there a number of variables being measured?

-- The measures mentioned under Results are different from that in Section 2.3 (sense of belonging, socio-educational inclusion, religiosity, discrimination)? The same variables should be listed under both Measures and Results.

-- Tables 1-4 have a list of questions and it is not clear whether these relate to the four Result headings or are separate measures of their own. Some results seem to be discussed through Tables and some in the text (e.g. was there a question asking if the respondent wore a hijab? lines 305-307). Were respondent characteristics [lines 155-162] also considered (e.g. some questions seem to divide results by gender, lines 338-340).

--Overall, this study's value seems clear, but its findings are obscured by the lack of a central question, clear definitions for central variables, and clear specification of which variables are being related in its statistical description and analysis.

Comments on the Quality of English Language

Quality of English language is fine overall - may need some editing to ensure greater clarity in communication.

Author Response

LETTER TO REVIEWER 4:

- This article considers the relationship between religiosity and national belonging from the perspective of young Muslims in Spain. This is a significant topic of research given that mainstream notions of national belonging are often implicitly or explicitly based on ethno-religious exclusions. However, the research design for this study is unclear, especially in terms of definitions for its main variables and how relationships between variables was analyzed.

- What is the central research question for the study? Lines 113-119 list a series of questions and it is not clear if there is an overarching question for the study, or if the study is a series of sub-questions examining relationships between a number of variables (religiosity, discrimination, belonging)? The term 'integration' is mentioned in lines 107-108, but is not part of the research questions.

RESPONSE:

  • We have thoroughly reviewed the research questions, variables and outcomes. Following the reviewer's recommendations, we have reworked the research questions, making them clearer for the reader, and divided them into a central question (on the three central variables under study) and specific ones: “In this study we address the following central research question: what are the perceptions of young Muslims…” (lines 121-130).

Also we have deleted the term "integration"

  • In order to provide greater clarity to the study, we detail in the "instrument" section the dimensions that make up the questionnaire, indicating which ones we focus on for this article (lines 179-191).

  • In the "measures" section, we clarify which are the central and secondary variables of the study (lines 207-215).

  • We also clarify some aspects of the variables and include the items of the questionnaire to which they refer. This can be seen in the "measurements" section.

  • In the table 1, we have also indicated in brackets to which variable of the questionnaire each item corresponds.

- It is mentioned that 'the study population consisted of a total of 2,091,656 Muslims' [line 127]. Is this the total number of Muslims between 18-24 years in Madrid, Andalusia, and Catalonia? It is then mentioned that 1,157 respondents were recruited for this study. This sample however is not a random sample since respondents were recruited through Muslim associations? The process of recruitment needs to be clarified further. It is fine if this is a convenience sample (which cannot be generalized to the wider population of Muslim youth in the three communities) since the study seeks to target respondents with specific characteristics.

RESPONSE:

  • We have modified the sampling technique used. As indicated by the reviewer, it is convenience sampling: “According to the demographic study of the Muslim community in Spain carried out by UCIDE (2020), a total of 2,091,656 Muslims were registered….” (lines 137-158).

- The measures mentioned in Section 2.3 need to be clearly defined. Are the questions mentioned in Section 2.3 different from the questions listed in Tables 1-4? What is the difference between religiosity and religious beliefs? Are socio-educational inclusion and sense of belonging part of the same measure?

RESPONSE:

  • We explain in more depth the variable "religious beliefs" and why it is a distinct variable from "religiosity".

  • We have modified the "results" sub-sections to make them clearer. For example: "feeling of belonging" has been changed to "socio-educational inclusion (feeling of belonging)". Since, as we indicated in the "measures" section, the feeling of belonging is included in the socio-educational inclusion dimension. Each "results" subsection corresponds to a central variable.

- Is religiosity the central variable? Are correlations being measured in terms of the relationship of religiosity to the other measures in Section 2.3? Or is belonging the central variable? Or are there a number of variables being measured?

RESPONSE:

The central variables, as indicated in the article, are religiosity, socio-educational inclusion and discrimination. We measure both the relationship and the significant differences between these variables and the "secondary" variables, which we have detailed in more detail in the "measures" section.

- The measures mentioned under Results are different from that in Section 2.3 (sense of belonging, socio-educational inclusion, religiosity, discrimination)? The same variables should be listed under both Measures and Results.

RESPONSE:

We have modified the subtitles of the results section, making it coincide with the central variables of the study that we explained in "measures".

- Tables 1-4 have a list of questions and it is not clear whether these relate to the four Result headings or are separate measures of their own. Some results seem to be discussed through Tables and some in the text (e.g. was there a question asking if the respondent wore a hijab? lines 305-307). Were respondent characteristics [lines 155-162] also considered (e.g. some questions seem to divide results by gender, lines 338-340).

RESPONSE:

We have clarified the items that make up the questionnaire in the "instrument" section. Likewise, in "measures" we have included the secondary measures, among which are the question of the veil, sex, generation, etc.

The tables can be better understood since the questionnaire items have been explained in both the "instrument" and "measures" sections.

To make it clearer, in Table 1 we have specified in parentheses to which scale of the questionnaire each item refers.

- Overall, this study's value seems clear, but its findings are obscured by the lack of a central question, clear definitions for central variables, and clear specification of which variables are being related in its statistical description and analysis.

RESPONSE:

We believe that the value of the article has improved following these revisions. We have included a central research question and secondary research questions. We have clarified in the "measures" section which are the main ones and which are the secondary ones. We have also defined them in greater detail and have incorporated the questionnaire items that make up each of them.

- Comments on the Quality of English Language

Quality of English language is fine overall - may need some editing to ensure greater clarity in communication.

RESPONSE:

We have reviewed the English language with a native translator.